# A Co-Created Assessment Framework to Measure Inclusive Health and Wellbeing in a Vulnerable Context in the South of Europe

**DOI:** 10.3390/ijerph21040510

**Published:** 2024-04-20

**Authors:** Isotta Mac Fadden, Roberta Cocchioni, María Mar Delgado-Serrano

**Affiliations:** 1Department of Agriculture Economics, Universidad de Córdoba, E-14005 Córdoba, Spain; isotta.macfadden@uco.es; 2IsIMPACT, 04100 Latina, Italy; robertacocchioni@gmail.com; 3WEARE Research Group, Department of Agriculture Economics, Universidad de Córdoba, E-14005 Córdoba, Spain

**Keywords:** inclusive health and wellbeing, community engagement, indicator co-creation, participatory assessment, neighbourhood context

## Abstract

Rapid urbanisation exacerbates health and wellbeing disparities in vulnerable contexts and underscores the imperative need to develop innovative and participatory co-creation approaches to understand and address the specificities of these contexts. This paper presents a method to develop an assessment framework that integrates top-down dimensions with bottom-up perspectives to monitor the impact of inclusive health and wellbeing interventions tailored to the neighbourhood’s needs in Las Palmeras, a vulnerable neighbourhood in Cordoba (Spain). Drawing upon studies in the literature examining urban health and wellbeing trends, it delineates a participatory and inclusive framework, emphasising the need for context-specific indicators and assessment tools. Involving diverse stakeholders, including residents and professionals, it enriches the process and identifies key indicators and assessment methods. This approach provides valuable insights for managing innovative solutions, aligning them with local expectations, and measuring their impact. It contributes to the discourse on inclusive urban health by advocating for participatory, context-specific strategies and interdisciplinary collaboration. While not universally applicable, the framework offers a model for health assessment in vulnerable contexts, encouraging further development of community-based tools for promoting inclusive wellbeing.

## 1. Introduction

The rise in urbanisation has had positive economic and social benefits but also a profound impact on health and wellbeing, increasing segregation and disparities and calling for the re-evaluation of strategies to ensure inclusivity and equity in urban settings [1,2]. The increasing number of urban residents, due to work opportunities, the better quality of services (healthcare, education, and transportation), and cultural diversity, intensify the challenges in providing health and wellbeing opportunities [3,4].

Health and wellbeing are interrelated concepts, and it is challenging to find a common, integrated definition since they are deeply personal concepts that everyone experiences in different ways at various stages during their life [5]. As stated in the preamble of the WHO Constitution of 1946, health is a state of complete physical, mental, and social wellbeing and not merely the absence of disease or infirmity. The enjoyment of the highest attainable standard of health is one of the fundamental rights of every human being, without distinction of race, religion, political beliefs, or economic and social conditions. Health promotion results from integrated actions encompassing economic, social, and environmental factors that empower people and communities to gain control and responsibility for their health [6]. There is no consensus around a unified definition of wellbeing. Still, there is general agreement that, as a minimum, wellbeing includes the presence of positive emotions and moods (e.g., contentment and happiness), satisfaction with life, fulfilment and positive functioning, and the absence or low presence of negative emotions like depression and anxiety [7,8] Wellbeing is a subjective and relative rather than an absolute concept, and reference points for judging wellbeing are individual hopes and ambitions informed by both objective circumstances and subjective perspectives [9].

The approach to inclusive health emerged to address the increasing health inequity and the urgent need to tackle disparities and promote equal access to healthcare services for all urban residents, especially in vulnerable and underserved areas [10]. It stresses the importance of not only focusing on individuals but also on health status at the community level [11]. The concept of inclusive health and wellbeing in cities has significantly evolved over the past century, shaped by changing societal attitudes, scientific advancements, and policy shifts. The focus on controlling infectious diseases and improving sanitation has shifted to a holistic approach that recognises the interconnectedness of physical, mental, and social wellbeing in urban spaces and demands a multifaceted approach to create sustainable and inclusive cities [12]. This paradigm shift suggests a fundamental change in how urban health is perceived and addressed, moving from traditional, siloed approaches to a comprehensive understanding of the factors influencing urban health and wellbeing [13].

Due to the inclusive approach and focus on communities, local policies are essential to promote, prevent, and mitigate socioeconomic inequalities in access to health and wellbeing among urban dwellers [2,14] and to boost inclusive health and wellbeing, including inhabitants’ perspectives and specific contextual needs [15]. Making cities more liveable, resilient, sustainable, and supportive of health and wellbeing is a challenge that involves several actors: urban planning and public health professionals, policy sectors, researchers, and practitioners, as well as the active participation of residents [16].

However, effective planning requires a transparent and accountable system of assessment based on sampling methods and indicators at the urban scale that is adapted to local contexts. The last decade has seen a growing emphasis on integrative and holistic approaches to sampling and measuring health and wellbeing [17] and an emphasis on the creation of indicators and assessment frameworks to monitor progress in health and wellbeing and promote its fair distribution [18,19]. Frameworks and indicators are essential tools for evaluating the impact of urban health initiatives, identifying gaps, and guiding evidence-based decision making [20].

In the context of inclusive health and wellbeing, these frameworks need to encompass a broad spectrum of factors, including social, economic, environmental, and cultural factors, that contribute to the overall health outcomes of urban dwellers [21]. By establishing clear parameters and measurable indicators, policymakers and stakeholders can effectively monitor the progress of interventions and assess their impact on different segments of the urban population. The standardised use of indicators enables comparative analyses across different cities, fostering knowledge sharing and best-practice dissemination to promote effective urban health policies globally [22]. Different indexes and indicators have been created and periodically calculated by relevant organisations and institutions: the WHO Indicators [23], the Human Development Index [24], the OECD Better Life Index [25] the World Happiness Index [26] and the Health Inclusivity Index [24]. Despite this change of perspective, it is hard to find a common framework [27], and research is still focused on some dimensions [28]. Furthermore, most of these indexes and indicators are proposed by experts and calculated at the country level, which hides the deep disparities at lower spatial levels. 

Comparable data on health and wellbeing at the city level are scarce and even more so at the community level. This is particularly true for small and medium-sized cities (SMSCs) that are home to most of the European population, where urbanisation is occurring at a faster pace [26] and where there is a pressing need for appropriate frameworks to measure health and wellbeing. Furthermore, there has been a noticeable increase in citizens’ interest in participating in the co-management of urban green spaces, habitats, and safety initiatives 5. This shift towards civic participation underscores the importance of involving community members in shaping urban policies and initiatives aimed at promoting health and wellbeing.

This paper’s objective is to display the process and the results of co-creating a participatory impact assessment framework for inclusive health and wellbeing in Las Palmeras, a vulnerable neighbourhood in Cordoba (Spain). Analysing health and wellbeing requires comprehensive frameworks that consider many components and how their interrelations shape people’s lives [27]. This approach encompasses the complex interrelationships among economic, psychological, social, and relational dimensions of health and wellbeing within this specific context and combines expert knowledge and local perspectives [29]. The framework is based on combining top-down dimensions and subdimensions influencing citizens’ health and wellbeing with the bottom-up perspectives of local inhabitants. The co-created indicators aim to monitor the impact of specific actions, aligning the current trends in inclusive health and wellbeing with the preferences and needs of neighbours. The research is part of the European H2020 project IN-HABIT, a five-year project whose objective is to investigate how inclusive health and wellbeing can be boosted through the co-design and the co-deployment of innovative actions—so-called visionary and integrated solutions (VISs)—based on aspects such as culture, art, food, human–pet relationships, and re-naturalization in four European cities located in the periphery of the European Union (Cordoba, Lucca, Riga, and Nitra). The focus is on peripheral SMSCs facing social challenges related to social conflicts and fragmentation, economic crises, lack of resources and skills, migrant flows and integration, and low access to services and opportunities.

IN-HABIT aims to advance knowledge on SMSCs’ health and wellbeing needs, define frameworks for collecting data at the city level, and elaborate data to monitor both the city-level evolutionary trajectories and the impact of interventions. The results will enhance the understanding of how SMSCs work in practice. A common working method was designed for the four cities, and in the second stage, the initial proposal was adapted to the specific city contexts. The proposed indicators aim to assess the impact of the initiatives implemented in every city regarding changes affecting the mental wellbeing, socioeconomic wellbeing, and healthy lifestyles of the urban dwellers in the project intervention areas. This paper presents the framework elaborated for Las Palmeras, a vulnerable neighbourhood in Cordoba, where the project will co-deploy different VISs, specifically focusing on the role of culture in enhancing health and wellbeing. Other researchers have previously suggested co-created frameworks and participatory approaches for assessing health and wellbeing [30,31], with a focus on physical health [32,33], or examining the use of public spaces for promoting health and wellbeing [34]. While ref. [29] proposed the development of indicators at a neighbourhood level, the authors’ emphasis was not specifically on vulnerable neighbourhoods. Additionally, their work primarily involved health authorities and concentrated more on physical health indicators. The novelty of our approach lies in the co-creation of health and wellbeing concepts as inclusive, complex, multidimensional, and context-specific and in the integration of top-down indicators with bottom-up views in the design of health and wellbeing indicators for vulnerable contexts, using a gender, diversity, equity, and inclusion perspective. This approach aims to capture the nuanced and varied experiences of individuals and communities in vulnerable settings, recognising that wellbeing encompasses more than just physical health. By incorporating diverse voices and perspectives, our framework offers a more holistic understanding of health and wellbeing in these contexts, providing valuable insights for targeted interventions and policy development.

## 2. Inclusive Health and Wellbeing Approach

The proposed multidimensional assessment framework is based on three core and interlinked theoretical constructs that will be described in what follows: (1) the social determinants of health theory; (2) the hedonic and eudaimonic theoretical approaches to subjective wellbeing; and (3) the gender, diversity, inclusion, and equity (GDEI) perspective that highlights the relationship between wellbeing and discrimination.

The social determinants of health describe the non-medical factors influencing health outcomes, including social wellbeing, economic wellbeing, spatial–environmental wellbeing, and healthy lifestyles [35]. These determinants integrate the conditions of birth, growth, work, and living, incorporating age and the broader array of forces and systems that configure daily life needs. These factors fall into individual, socioeconomic, and environmental areas [36] and encompass economic policies and procedures, development agendas, social norms, social policies, and political systems [37]. Health and illness follow a social gradient: the lower the socioeconomic position of the person, the worse the health status. The most important structural stratifiers and their proxy indicators include income, education, occupation, social class, gender, and race/ethnicity. Health and wellbeing are influenced not only by individual attributes but also by the social circumstances in which persons find themselves and the environments in which they live; these determinants interact with each other dynamically and may threaten or protect an individual’s mental health state. Furthermore, the underlying social determinants of health inequities operate through intermediary determinants, such as material circumstances, psychosocial conditions, behavioural and biological factors, and the health system itself, shaping overall health outcomes [6].

Healthy lifestyles and behaviours also positively affect health and wellbeing [38]. A healthy lifestyle is a way of living that lowers the risk of being seriously ill or dying early and includes aspects such as eating, exercising, and having an active life, with no additions, enough sleep, recreation, and mind exercising [23]. There is no optimal lifestyle to be prescribed for all people, but both inequalities and discrimination have a deep impact on people’s health and lifestyles. Personal characteristics such as “culture, income, family structure, age, physical ability, home, and work environment will make certain ways and conditions of living more attractive, feasible and appropriate” [39]. The “individualist paradigm” in health lifestyle [40] focused on attitudes and motivational structures at the individual level and prevailed in medical research analyses in past decades. However, more recently, it has been considered too reductive, as it overlooks the complexities of social action. Lifestyles are not solely about individual choices and will for self-representation but are also shaped by contextual factors embedding individual choices in social contexts [41]. A healthy lifestyle can also be a positive role model for family or community members, particularly children [23]. Therefore, healthy lifestyles might be defined as a way of living that integrates an individual’s personal characteristics with social interactions and socioeconomic and environmental living conditions [42].

Subjective wellbeing refers to an individual’s own assessment of their quality of life and their situation and, as such, is a multifaceted construct that brings together the hedonic research approach, which stresses constructs such as happiness, positive affect, low negative affect, and satisfaction with life [43], and the eudaimonic tradition, which highlights positive psychological functioning and human development [44,45].

The construct consists mainly of two main components: positive aspects of subjective wellbeing, as measured by general psychological wellbeing, which contains components of positive emotions, physical wellbeing, interest [46], and life satisfaction [45]; and negative aspects of subjective wellbeing, including mental distress and poor mental health [47]. Subjective wellbeing includes people’s appraisals and evaluations of their own lives from reflective cognitive judgements, such as life satisfaction, and emotional responses to ongoing life in terms of positive and pleasant emotions and unpleasant and negative emotions [48].

The inclusive approach to health and wellbeing incorporates a cross-cutting GDEI perspective, emphasising the relationship between wellbeing and discrimination. Discrimination, often stemming from prejudiced attitudes, disempowers individuals, hinders their active participation, restricts skill development, and often obstructs access to essential opportunities, such as work, health services, education, and housing [49]. Consequently, it directly affects the targeted individuals and groups while exerting indirect and profound consequences on society. A society permitting or tolerating discrimination restricts individuals’ abilities to freely realise their full potential, both for themselves and for society [50]. Perceived discrimination is recognised to have impacts on both physical and mental health. Although extensive research has explored the links between mental health and those vulnerable to exclusion and discrimination, the impact of other forms of discrimination on health remains understudied [51].

Minority stress represents an additional layer of stressors experienced by minority groups. Those belonging to minority groups face an extra burden of stress that requires adaptive efforts beyond what is experienced by non-minorities. Meyer [52] formulated the minority stress model based on his research into the mental health of LGBTQI+ individuals. This model outlines the relationships between social stressors and the mental health of these individuals, identifying the mechanisms through which social stressors impact the health and wellbeing of this community and the harm that prejudice and stigma cause.

## 3. Methods

### 3.1. Study Area

Cordoba is a medium-sized city in the South of Spain with a population of 323.763 inhabitants [53] and a unique historical and cultural heritage, evidenced by its four UNESCO World Heritage Sites. Despite this rich cultural backdrop, the city faces significant socioeconomic challenges, including a high unemployment rate (28.46%) and the presence of 5 out of the 15 most marginal neighbourhoods in Spain, among them Las Palmeras. Las Palmeras is a small neighbourhood of 2212 inhabitants [53] located on the outskirts of Cordoba and characterised by segregation and disconnection (both from the city and internally among its inhabitants), high dependence on social subsidies, unstructured families and gender violence, absence of role models, failure of educational models, robberies, drug trafficking, illegal activities, and police raids. Health and wellbeing levels are well below the city’s standards. The health status is characterised by unhealthy diets and lifestyles, obesity problems [54], unwanted pregnancies, and drug consumption from early ages. Wellbeing is limited by the lack of employment, the low quality of social houses, the absence of incomes to afford minimum welfare, the lack of green areas and public spaces, the low educational standards, and insecurity due to illegal activities. People have limited feelings of belonging or identity. Collective and community actions are almost non-existent. Being born in Las Palmeras is a stigma that makes many people hide their origins or the place where they live and drives them to leave the neighbourhood once they are better off. Different inclusion strategies have been tested over the years (vocational training courses, skill censuses to offer jobs, and participation in city events) [55]. Still, these were short-term, top-down, fragmented, and isolated initiatives that did not have continuity and achieved limited results, leading to scepticism regarding social transformation and better health and wellbeing opportunities in the neighbourhood. The neighbourhood faces what the authors of [56] consider cyclical vulnerability exacerbated by the absence of integrated perspectives in social policies and reliance on fragmented approaches to social resources and subsidies.

### 3.2. Top-Down Measures of Health and Wellbeing

The selection of top-down dimensions and subdimensions for inclusive health and wellbeing (Table 1) is grounded in the theoretical approach detailed in Section 2 and a review of validated frameworks at both European and international levels. The selection process involved examining official websites and documents of influential entities globally acknowledged for their impact in defining and measuring health and wellbeing, such as the United Nations, the World Health Organization [23], the Organisation for Economic Co-operation and Development [25], and European Agencies. The research team, comprised of three local researchers, four experts from ISIMPACT (an impact assessment company), and three researchers from Reading University, systematically explored relevant frameworks and indexes related to health and wellbeing. This exploration aimed to identify and validate a preliminary set of dimensions and subdimensions of health and wellbeing. Drawing on this thorough review and their extensive experience in vulnerable urban contexts and inclusive perspectives, the researchers proposed key aspects essential for assessing the impact on inclusive health and wellbeing resulting from the VISs to be implemented by the IN-HABIT project.

### 3.3. Bottom-Up Identification of Local Perceptions on Health and Wellbeing with GDEI Perspective

This phase consisted of a series of meetings, interviews, and questionnaires involving inhabitants and representatives of local target groups in Las Palmeras, including people at risk of discrimination and exclusion, to unveil their perceptions of health and wellbeing, the aspects and dimensions that they consider, and the impacts they expect from the project. Other aspects identified were the target groups of the actions and the psychosocial risk factors associated with the conditions of the local community. A targeted engagement strategy was employed which involved participant observation and informal discussions with residents and representatives from various public and private institutions to identify potential participants. Our initial efforts aimed to identify key formal and informal entities within Las Palmeras, such as social services, associations, NGOs, sports clubs, and religious charities. Subsequently, our focus shifted towards the most vulnerable segments of the community. To gain a comprehensive understanding, we conducted extensive research on minority and marginalised groups, including door-to-door visits to all 700 apartments in the neighbourhood, most of which attended to us and helped us to better understand the reality of the inhabitants.

Twenty-six face-to-face semi-structured interviews with local stakeholders and inhabitants were conducted. The respondents were neighbours and representatives of the described civic associations, as well as private companies working in the neighbourhood. They were selected to ensure diversity by age, disability, ethnic origin, gender identity, and sexual orientation, as well as intersectionality. The intersectional approach ensures the representation of those facing multiple challenges, such as women dealing with gender discrimination and violence living in a stigmatised neighbourhood, as well as elders, late adolescents, or unemployed women, who are considered among the most vulnerable. We also interviewed individuals who have successfully navigated similar circumstances and now have employment or are young university students. Additionally, perspectives from relevant professionals were included. This comprehensive approach deepened our understanding of the challenges faced by vulnerable populations. Seventeen respondents lived in the neighbourhood, nine of whom have worked on social projects in the area for more than five years. The interviews lasted around two hours each and were conducted in a comfortable environment selected by the person interviewed. Informed consent had been previously signed. The interviews were audio-recorded using a device owned by the University of Cordoba (UCO), and the recordings were transcribed.

The interviews were focused on (i) identification of the most relevant dimensions of health and wellbeing; (ii) identification of the most significant changes that the project could produce regarding local people’s health and wellbeing (in general and for specific groups at risk of discrimination and exclusion); (iii) effects of the COVID-19 pandemic on inhabitants’ health and wellbeing; (iv) positive and negative aspects of living in Las Palmeras; and (v) perception of the role of culture and heritage in boosting health and wellbeing. A systematic analysis was used to find patterned responses or topics in the narrative set connecting inclusive health and wellbeing and specific dimensions in the vulnerable context of Las Palmeras, applying thematic analysis techniques [63]. The UCO researchers conducted the initial analysis. Then, the company experts independently validated the emerging themes by examining the data and merging them with their analytic contributions, identifying, analysing, and reporting patterns, themes, and subthemes using a constant comparative analysis [64]. After several readings to ensure familiarity with the content and the narratives, a coding process was developed by three researchers to organise data into meaningful groups that linked the top-down dimensions and subdimensions to the aspects and concepts mentioned by Las Palmeras participants. Three rounds of coding were performed to ensure rigorous analysis using Microsoft Excel tables. All team members have expertise in socioeconomic studies and qualitative research methods, and the local researchers also know the context well.

A questionnaire was passed to representatives of organisations working in the neighbourhood to include the perspectives of people with personal characteristics related to GDEI. The survey included items to profile the characteristics of the organisations represented and their missions and visions and, more specifically, to collect the interviewees’ perceptions on health and wellbeing in vulnerable contexts, as well as the expected contributions of IN-HABIT to vulnerable groups, the attitudes towards discriminated groups, the most common situations of social exclusion, and the aspects of life and environment that most affect the health and wellbeing of the target population. Thirteen questionnaires were answered by organisations representing/working with ethnic minorities (five representatives), people at risk of social exclusion (three representatives), young people at risk of social exclusion (one representative), religious minorities (one representative), minors at school age (one representative), elderly people (one representative), and families with children (one representative). Relevant aspects that characterise health and wellbeing for people in vulnerable contexts and at risk of discrimination emerged from this exercise. At the same time, this analysis was used to identify the areas of intervention that might contribute more to boosting inclusive health and wellbeing for these target groups.

### 3.4. Final Set of Inclusive Health and Wellbeing Indicators

The process of selecting dimensions, subdimensions, and indicators for inclusive health and wellbeing involved the combination of top-down and bottom-up perspectives to harmonise indicators and scales with insights from residents and representatives of community organisations. This inclusive approach facilitated the identification of key aspects to evaluate the impact of project initiatives. The bottom-up analysis was particularly instrumental in aligning the project VISs with the specific needs of the local inhabitants. The methodology followed a circular process, promoting mutual learning between researchers and community members. Results from each phase were used to revise and validate preceding stages and contributed to accumulating new knowledge regarding the local context and expectations.

The results of the bottom-up and the top-down explorations were used to formulate a set of specific indicators to assess the impact of the project VISs in Las Palmeras. The perceptions of the local people and the different aspects they mentioned were translated into indicators. Most of the indicators were linked to expected changes in health and wellbeing that could be attributable (at least partially) to the project interventions, and others were context indicators to characterise the situation. They were decided after three rounds of consultation and consensus generation to meet the following criteria: (1) comparability: the indicators should be able to measure those aspects of health and wellbeing considered by the main European and international statistical and research frameworks (WHO, OECD, Eurostat, Eurofound, European Commission, UNDP/SDGs) to ensure comparability and to fill the gaps in terms of data availability for SMSCs; (2) specificity: the indicators should be meaningful and relevant to the specific local context since they refer to existing and measurable characteristics of the local population, project solutions, and socioeconomic and institutional contexts, and should consider the effects of the COVID-19 pandemic; and (3) inclusiveness: the subjective indicators consider both the researchers’ and the inhabitants’ assumptions about expected changes affecting health and wellbeing, with specific regard to the perspectives of those people who identified themselves as representatives of the groups with GDEI personal characteristics at the local level [65].

## 4. Results

### 4.1. Top-Down Dimensions and Subdimensions of Health and Wellbeing Results

After analysing internationally validated frameworks, the researchers proposed a first set of dimensions and subdimensions of inclusive health and wellbeing to be included in the framework (Figure 1).

### 4.2. Bottom-Up Aspects Associated with Health and Wellbeing in Las Palmeras

The interviews and questionnaires revealed key bottom-up dimensions of health and wellbeing (Table 2), adding a resident-centred approach. Through a qualitative analysis, the added value of the inhabitants’ perspectives was incorporated, enriching the meaning of health and wellbeing in this context [66]. Participants consistently expressed a global perspective, emphasising the importance of physical, psychological, and spiritual fulfilment. This encompassed emotional, economic, family, environmental, and community dimensions, as illustrated by Participant Four, who emphasised the significance of “being well with all that surrounds me, with nature”. The multifaceted aspects included safety, relational networks, physical health, and adherence to healthy lifestyles, such as good nutrition, hygiene, and quality sleep, due to their impact on mood and school attendance. Participants underscored the necessity of healthy leisure alternatives, suitable spaces, and facilities, particularly emphasising the demand for public and community spaces to nurture a healthy environment. From the inhabitants’ viewpoint, promoting health and wellbeing in a vulnerable context requires ensuring basic needs (economic, cultural, and educational) with stable sources of income to alleviate stress and insecurity.

Additionally, they mentioned the quality of coexistence between neighbours, absence of anxiety, inner peace, harmony with the environment, and mental openness to diversity as influential factors. This openness contributes to a sense of belonging and identity with the city, evading the feeling of living in a stigmatised ghetto. Participants also believed that the IN-HABIT project could significantly impact the neighbourhood by fostering inclusive health and wellbeing. Their demands are focused on creating healthier, greener, and more inclusive spaces, promoting healthy habits and leisure activities, and raising awareness about gender roles and inequity. The role of culture was barely mentioned because it did not have a direct relation to the health and wellbeing of the participants.

### 4.3. Final Assessment Framework

The final list of inclusive health and wellbeing indicators for the Cordoba case study (Table 3) resulted from the integration of the top-down approach that drew on dimensions, subdimensions, and indicators found in the literature and the bottom-up process that identified the perceptions and most-valued aspects of health and wellbeing for the participants. The final set of indicators includes proposed metrics and references the anticipated changes they are designed to measure. This dual approach incorporates both global perspectives and locally valued dimensions. While most indicators aimed to measure changes, some served as context indicators, contributing to a more nuanced characterisation of the intervention area, including the indicators to measure subjective wellbeing based on validated international scales. The indicators for Las Palmeras were supplemented with some shared indicators for the four cities, covering sociodemographic aspects, GDEI characteristics, and the impact of COVID-19 on healthy lifestyles. The identified indicators will be crucial in evaluating the impact on specific dimensions and subdimensions of inclusive health and wellbeing resulting from the VISs deployed by the project. Most of the indicators will be measured through surveys and questionnaires due to their qualitative nature. An initial step in this process involved the development of a baseline survey based on questions to measure these indicators. The survey was reviewed by the School Research Ethics Committee of the University of Reading (UK) (project 2021-085-RM) and served as a foundation to compare future assessments. Table A1 in Appendix A presents a description of each indicator.

## 5. Discussion

The understanding of health and wellbeing has evolved beyond individual characteristics to acknowledge the profound influence of contextual factors. Growing inequities have prompted global attention to developing participatory urban health indicators that capture the unique attributes of specific areas [64]. These indicators serve as a foundation for directing efforts towards promoting good practices and measuring progress, particularly in vulnerable contexts [4]. The complexity of health and wellbeing extends beyond conventional metrics like income or growth levels, encompassing elements such as access to public services and the quality of community social and political life, which traditional indicators may not directly measure [67]. Adopting comprehensive and participatory approaches means recognising that inclusive health outcomes result from multiple causative factors within contexts marked by broader vulnerabilities and that dweller perception is considered in the interventions and their measurement. A neighbourhood, for instance, can be viewed as a social determinant of health, impacting functional wellbeing through aspects like education, employment, housing quality, and the availability of green spaces. Living in a vulnerable context inherently increases the risk of experiencing poor health and wellbeing. Consequently, including specific dimensions increases relevance and significance when designing indicators for vulnerable contexts [68].

### 5.1. The Co-Creation Process

While the theoretical debate on the importance of urban policies in enhancing health and wellbeing is well-established [69], a robust body of literature on practical approaches to boost health and wellbeing and measure their impact on inhabitants, especially in vulnerable communities and based on people’s perceptions, is lacking [70]. Despite an extensive literature emphasising the added value of participatory research, there is still a lack of methodologies and tools to implement it. The complexity deepens when aiming to understand the dimensions of health and wellbeing by integrating the perspective of vulnerable populations and vulnerable contexts [71].

The study’s uniqueness lies in the integrated co-design approach of the impact assessment, in which a co-created assessment framework has been developed integrating top-down and bottom-up approaches and including different stakeholders, such as local inhabitants and NGOs and entities working in a vulnerable context [72]. The top-down approach provides understanding and knowledge of the relevant research topics, an objective and holistic vision, and awareness of factors that are indirectly involved (cause–effect relationships) [73]. However, the wide range of factors shaping health and wellbeing in a specific context cannot be understood without including the vision, the culture, and the social aspects of the community [74]. The bottom-up approach provides in-depth and specific knowledge of problems through direct experience and is rooted in social history and context dynamics. Nevertheless, it might lack expertise and previous knowledge in the field of research and offer a subjective vision bounded by history and a partial interpretation of problems. Additionally, bottom-up processes are always limited by the level of representativeness as it is difficult to ensure a sufficiently representative sample that includes the diversity of views and not to fall into the danger of being overpowered or offering a fragmented vision of the context conditioned by the willingness/capacity for active participation [75].

An integrated approach, if matched by a deep knowledge of the reality, experience, and professionalism of researchers so as to avoid superficial conclusions that reproduce stereotypes related to vulnerable contexts, might be the best option to cover the limitations of both approaches and offer an in-depth, comprehensive understanding of the dimensions. In this research, the combination of both approaches and the further building up of a consensus have guided the final selection of indicators. The co-creation approach made it possible to add valuable expertise and knowledge driven by an in-depth comprehension of the complexity of the context. The circular learning process and the participants’ feedback have supported the inclusion and validation of indicators considered significant for all the participants, combining local cultural context expertise and scientific knowledge on the main aspects that need to be considered to boost inclusive health and wellbeing [76].

### 5.2. The Integration of Top-Down and Bottom-Up Perspectives

In developing frameworks for inclusive health and wellbeing in cities, it is crucial to incorporate both top-down and bottom-up approaches to ensure a comprehensive understanding of community needs and aspirations [77]. While top-down approaches are essential for establishing overarching guidelines and regulatory frameworks, they may overlook diverse urban communities’ nuanced realities and lived experiences. In contrast, bottom-up approaches, driven by community engagement, participatory research, and grassroots knowledge, provide valuable insights into the challenges and priorities shaping health and wellbeing at the neighbourhood level [29]. By actively involving community members, local organisations, and marginalised groups in designing and implementing assessment frameworks, we gained a deeper understanding of the contextual nuances and social determinants influencing health and wellbeing outcomes in Las Palmeras. This inclusive approach ensures that the resulting frameworks are more accurate, relevant, and responsive to the community’s specific needs.

In vulnerable contexts and among collectives at risk of social exclusion, the needs for health and wellbeing are often multifaceted and require targeted, tailored approaches to address the underlying determinants of poor health outcomes [2]. Understanding these communities’ specific challenges is crucial for designing effective interventions that promote inclusivity, equity, and resilience [78]. Here, we delve deeper into the unique health and wellbeing requirements in vulnerable contexts and for collectives at risk of social exclusion. By aligning project interventions with community expectations, they can become more responsive and tailored to the diverse needs of the people, thereby enhancing the effectiveness and sustainability of such interventions. Moreover, community engagement promotes a sense of ownership and empowerment among residents and fosters social cohesion [68].

In vulnerable contexts, a wider range of multicausal dimensions need to be assessed to boost health and wellbeing [14]. The biggest obstacles perceived in our case study are the strong sense of discrimination and stigma, uncovered basic needs, unemployment, education, and insecurity due to high levels of violence. Important aspects of enhancing the health and wellbeing of Las Palmeras’ inhabitants were creating job opportunities, increasing self-esteem, deconstructing stigma, increasing the sense of safety, generating healthy habits, and creating green spaces for socialisation and healthy activities. The process evidenced that neighbours are aware of the importance of living in a healthier environment, having healthy habits, and feeling safe in public and green spaces without the threat of violence [79]. Other relevant aspects in our context were the role of empowerment in self-esteem and openness to change (new opportunities) [80]. Some specific indicators that emerged from this analysis were awareness of and motivation towards healthy habits, satisfaction with personal relationships in the neighbourhood, trust in others, and openness to diversity. The limited access to culture and cultural activities meant that the local people barely mentioned these aspects. This is explained by the lack of options for culture or leisure in Las Palmeras and the fact that the basic needs for health promotion and changing behaviour are often not covered in disadvantaged areas [81]. However, due to their role in the project, indicators linked to them have been introduced. Finally, the in-depth interviews allowed us to identify the undervalued resources that the community possesses and valorises, such as strong resilience, the relevance of traditions and cultural heritage, and a strong sense of community. These aspects might be of great interest to boost changes that improve health and wellbeing.

## 6. Conclusions

The primary contribution of this research lies in establishing a working method to assess health and wellbeing in vulnerable contexts through inclusive and participatory approaches tailored to the specificity of each context and its unique needs. By engaging a diverse range of stakeholders with varying levels of knowledge about the context, we identified the most necessary VISs and co-designed their assessment methods. This inclusive approach, incorporating the perspectives of both local residents and professionals, enriched the research experience.

Through this working method, we uncovered the factors and dimensions that shape health and wellbeing in the specific setting under study, providing valuable insights for managing and evaluating innovative solutions and assessing their replicability. Furthermore, by incorporating the perspectives of local inhabitants regarding the changes needed to promote inclusive health and wellbeing, we can align the project VISs with their expectations, thereby enhancing their effectiveness and relevance.

The joint work of researchers and local actors allowed cross-fertilisation and the recognition of the multidimensional and multilevel understanding of the complex elements that build the web of health and wellbeing in every context. The selection of dimensions and the definition of indicators have been essential steps in monitoring and assessing the impact of the VISs to be deployed, better allocating resources, and investigating the effectiveness of health policies without forgetting the specificity of each context. This highlights the role of science in supporting policymaking but also calls for active citizen involvement and empowerment in shaping policies and engaging with initiatives to explore problems, find solutions, and monitor their impact.

The assessment framework would enable the study of health and wellbeing with an inclusive perspective (e.g., through culturally adapted assessment tools) and may contribute to identifying the specific needs of vulnerable populations for targeted decisions, resource allocation, and policy development. The multidimensional health and wellbeing assessment adds significant value by considering factors beyond the traditional measures. The process described encourages further work in developing useful, meaningful, and sustainable community-level tools that can be used to assess and boost health and wellbeing with an inclusive perspective.

## 7. Limitations

Some limitations of the framework are the difficulties in generalising it to other contexts and the fact that most of the indicators are based on people’s perceptions. Our set of indicators remains contextual and corresponds to the knowledge level at a given moment in our specific neighbourhood, mainly measured through direct questioning methods. However, we provided a methodology to build new ones dynamically that can be replicated in vulnerable contexts and generate more adapted definitions of inclusive health and wellbeing and assessment methods. Furthermore, the list of indicators can inspire the work of others.

The limited number of representatives in some of the bottom-up categories could be considered another shortcoming, but we involved all the institutions working in the area. Finally, the fact that the co-design depended on the actors’ capacities can be mentioned. Complex hierarchical structures, low literacy, limited participation skills, or traditional local values can limit findings in other contexts.

## Figures and Tables

**Figure 1 ijerph-21-00510-f001:**
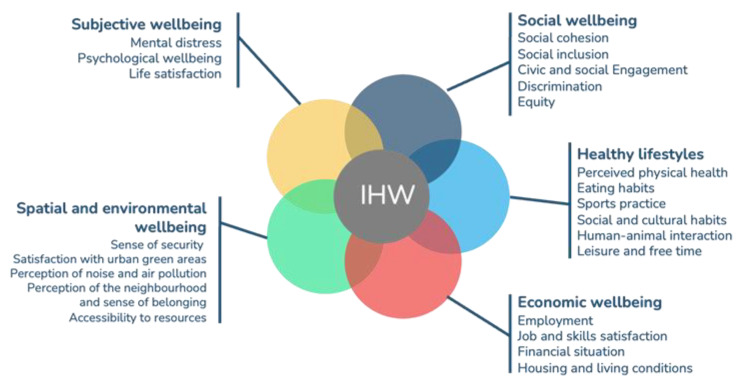
Top-down dimensions and sub-dimensions of health and wellbeing (dimensions are in bolt and larger font size).

**Table 1 ijerph-21-00510-t001:** Frameworks analysing health and wellbeing at the European and international levels.

European/International Organisations	Assessment Frameworks
United Nations	Human Development Index [57] Sustainable Development Goals (SDGs)
WHO—World Health Organization	Measuring the Quality of Life (WHO-QOL) WHO Five Wellbeing Index (WHO-5) [58]
OECD—Organisation for Economic Co-operation and Development	OECD Better Life Initiative. Your Better Life Index [25]
Eurostat	Final report of the expert group on quality-of-life indicators [59] European Union Statistics on Income and Living Conditions Sustainable development in the European Union. Monitoring report on progress towards the SDGs in an EU context [60]
EUROFOUND—European Foundation for the Improvement of Living and Working Conditions	Living conditions, social exclusion, and mental wellbeing Employment security and employment: A contribution to the flexicurity debate
European Environmental Agency	Healthy environment, healthy lives: how the environment influences health and wellbeing in Europe [61]
Horizon 2020 EKLIPSE Project	An impact evaluation framework to support planning and evaluation of nature-based solutions projects [62]

**Table 2 ijerph-21-00510-t002:** Local perceptions of health and wellbeing in Las Palmeras (participants’ comments are in italics and the number of times the items were mentioned in the interviews are in brackets).

Dimensions	Aspects Mentioned in the Narrative
Holistic perspective on health and wellbeing	Physical, psychological, emotional, spiritual, and economic stability (3) Community health and a secure environment (“at home and in the neighbourhood”), good coexistence and family life (3) Life-long education (with an intergenerational perspective) and social skills (1)
Basic needs satisfaction	Living with tranquillity and without stress to cover basic or economic needs (1) Stable and secure sources of income (“there are families with four children living on social benefits of 400 euros”) (5) Having basic needs covered (4) Living with dignity means having a job (3) Guaranteed basic needs, including economic, cultural, and educational needs (1) Having access to food (1) Living in an adequate space (“often several generations share households and live in small rooms with negative consequences for individual development”) (1) Clean and decent housing (1) Having a roof to live under (1)
Environment	Feeling safe and secure (7) Having a silent environment to be able to sleep, study, etc. (3) A healthy and secure environment (2) Feeling good about what surrounds oneself (2) A well-conserved environment (“I can take care of my house, but if my environment is not adequate…?”) (1) Environmental protection and preservation (1) Contact with nature (1) Caring for the neighbourhood infrastructures (1)
Healthy habits/physical health	Physical health (4) Healthy habits (2) Good-quality food (2) Lack of sleep (“affects mood, school attendance”) (1) Hygiene (1)
Healthy leisure	Being able to have fun and time for leisure (1) Pleasant and safe leisure spaces for all ages (5)
Psychological or mental wellbeing	No stress (2) Feeling well with yourself, self-fulfilled (academic/career success) (2) Being calm, feeling safe, and not having anxiety (1) Feeling happy (1) Being motivated and having goals (1)
Inclusiveness, equity, and equality	Sense of belonging and inclusion in the city (2) Feeling valued (1) Feeling loved (1) Personal growth (1) Feeling you have equal opportunities (“economic, educational, job”) (1)
Self-care and family care	Self-care (“lack of hygienic care is very serious in some families, there are posters in some of the social service offices to explain personal hygienic patterns”) (1) Care for my family and also for the house and the environment (1)

**Table 3 ijerph-21-00510-t003:** Assessment framework for Las Palmeras, including dimensions (Ds), subdimensions (SDs), indicators, and expected changes promoted by the VISs.

Ds	SDs	Expected Change	Indicator
SOCIAL WELLBEING	Social cohesion	Improved social relations	Satisfaction with personal relationships in the neighbourhood
		Increased trust among people	Trust in others
		Improved social network support	Social network support
	Social inclusion	Increased social relations in public spaces	Sense of inclusion
			Contact with others in public spaces
			Domestic isolation
		Improved openness to diversity	Openness to diversity
	Civic and social engagement	Improved social engagement	Engagement in voluntary activities
			Engagement in local community activities
			Engagement in caring for common spaces
		Improved civic engagement	Democratic participation
			Involvement in local policies
		Increased change-making attitude	Change-making attitude
	Equity	Context indicator	Sense of being treated with equity
		Improved equal access to culture and leisure	Access to culture and leisure
		Context indicator	Obstacles to access to culture and leisure
		Context indicator	Obstacles to access to social care services and health services
		Context indicator	Obstacles to access to training opportunities
		Context indicator	Access to the internet from home
	Discrimination	Context indicator	Perception of discrimination in society
		Context indicator	Perceived personal condition of discrimination
		Increased collective self-esteem	Collective self-esteem
SPATIAL AND ENVIRONMENTAL WELLBEING	Spatial wellbeing	Improved accessibility of local resources	Accessibility of local resources
		Improved satisfaction with urban green areas	Satisfaction with urban green areas
		Increased inclusiveness of public squares and green areas	Inclusiveness of public squares and green areas
		Improved sense of belonging and satisfaction with the quality of the neighbourhood	Positive perception of the neighbourhood
		Context indicator	Air pollution
	Perception of security	Increased sense of safety	Sense of safety at night
			Sense of safety in green areas
		Context indicator	Crime, violence, or vandalism in the living area
ECONOMIC WELLBEING	Employability	Increased employability	Opportunity to find a job
			Adequation of skills to the job market
			Options to find a job in the expected sector
		Increased satisfaction with job and skills	Job and skill satisfaction
	Financial situation	Context indicator	Basic needs satisfaction
		Context indicator	Financial situation satisfaction
		Context indicator	Time and resources for personal care satisfaction
		Increased satisfaction with environment	Surroundings/living environment satisfaction
HEALTHY LIFESTYLES	Physical health status	Context indicator	Self-reported health status
	Healthy food habits	Context indicator	Time spent on food preparation at home
		Increased consumption of self-grown fruit and vegetables	Self-grown fruit and vegetable consumption
		Context indicator	Consumption of fruits and vegetables
		Context indicator	Access to healthy and nutritious food
		Increased awareness of and motivation towards healthy habits	Awareness of and motivation towards healthy habits
	Sports practice	Context indicator	Practice of physical activity
		Increased practice of sports in public green areas	Practice of sports in public green areas
		Increased perception of benefits from sports	Benefits from sports
	Cultural consumption and production	Increased satisfaction with cultural facilities	Satisfaction with cultural facilities
		Context indicator	Cultural consumption
		Increased participation in cultural activities within public spaces	Participation in cultural activities within public spaces (outdoor/indoor)
		Increased perception of benefits from culture	Benefits from culture
		Increased local cultural engagement	Local cultural engagement
	Leisure/free time	Context indicator	Time devoted to leisure and personal care
		Increased practice of healthy leisure	Practice of healthy leisure
		Increased relaxation/exercising time in public green areas	Time spent playing or relaxing in public green areas
		Increased time in social and recreational public spaces	Time spent in social and recreational public spaces
		Context indicator	Time devoted to family care
		Context indicator	Time devoted to pets’ care/playing with pets
		Context indicator	Satisfaction with free-time use
		Increased perception of public spaces benefits	Benefits from social and recreational public spaces
SUBJECTIVE WELLBEING	General psychological wellbeing (positive emotions) WHO-5 Scale	Context indicator	Feeling cheerful and in good spirits
		Context indicator	Feeling calm and relaxed
		Context indicator	Feeling active and vigorous
		Context indicator	Feeling fresh and rested
		Context indicator	Feeling that one’s life has been filled with things that interest oneself
	Mental distress Kessler Psychological Distress Scale K6	Context indicator	Feeling nervous
		Context indicator	Feeling hopeless
		Context indicator	Feeling restless or fidgety
		Context indicator	Feeling depressed
		Context indicator	Feeling that everything is an effort
		Context indicator	Feeling worthless
	Life satisfaction Ryff’s life satisfaction scale	Context indicator	Life satisfaction

## Data Availability

According to the Data Management Plan of IN-HABIT approved by the EC, the datasets will be made available at the end of the project (August 2025) at the Zenodo community of the project https://zenodo.org/communities/in-habit-h2020/records?q=&l=list&p=1&s=10&sort=newest.

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
