# Peer review of "A Co-Created Assessment Framework to Measure Inclusive Health and Wellbeing in a Vulnerable Context in the South of Europe"

_ijerph, 2024, doi:10.3390/ijerph21040510_

Round 1

Reviewer 1 Report

Comments and Suggestions for Authors

Thank you for the opportunity to peer-review this important work that provides an example of top-down and bottom-up co-design. The Authors have also incorporated community responses to COVID-19 into the research to increase its generalisability/ transferability to future pandemics. Minor comments for consideration: (i) the use of abbreviation, noting that WHO was abbreviated and used before the term was written in full. Also, note that the Authors spelt "Organisation" with a -s rather than -z, which is the correct proper noun for the WHO. (ii) Would the authors comment that the participants in the qualitative phase of the project were recruited for and/or addressed all the relevant intersectionalities of the social determinants of health framework? The latter is to demonstrate the credibility and comprehensiveness of the extensive program of work.

Author Response

Dear Reviewer

Thanks for your nice comments on our work. We are very pleased that you found our work relevant now and in the future. Please see below our response to your comments

  • the use of abbreviation, noting that WHO was abbreviated and used before the term was written in full. Also, note that the Authors spelt "Organisation" with a -s rather than -z, which is the correct proper noun for the WHO.

Response: We appreciate you noticing this typo. As we used British English, we didn’t realise. Now, it has been corrected.

  • Would the authors comment that the participants in the qualitative phase of the project were recruited for and/or addressed all the relevant intersectionalities of the social determinants of health framework? The latter is to demonstrate the credibility and comprehensiveness of the extensive program of work.

Response: To address your suggestion, we have clarified it in lines 293-300, and it now reads as follows: "The intersectional approach ensured representation of those facing multiple challenges, such as women dealing with gender discrimination and violence living in a stigmatised neighbourhood, as well as elders, late adolescents or unemployed women who are considered among the most vulnerable. We also interviewed individuals who have successfully navigated similar circumstances and now have employment or are young university students. Additionally, perspectives from relevant professionals were included. This comprehensive approach deepened our understanding of the challenges faced by vulnerable populations."

Reviewer 2 Report

Comments and Suggestions for Authors

Thank you for the opportunity to have read this manuscript. I find it well written, well researched, and contains excellent insights for future studies on the improvement of at-risk communities. 

In my opinion, the strengths of the paper are the robustness of the literature review, the relevance of the topic, the fluent and focused writing, the appropriate methodology, and the rationale supporting the need to create a specific instrument to assess at-risk communities. Weaknesses have already been highlighted among the limitations of the article by the authors themselves, first and foremost the applicability of the tool to the context in which it was tested and the possible difficulty of applying it to other contexts without the necessary modifications. However, I think it can certainly be a good starting point for other studies on other disadvantaged communities.

I was wondering if there were any references for what is stated in section 3.1, specifically from line 220 to the end. I believe it would be helpful to provide references to support the statements, as well as some examples of failed inclusion strategies.

Author Response

Dear Reviewer

Thanks for your comments. We are very pleased to hear that you found our work interesting, addressing pressing topics and useful for the audience.

In relation to your comment on including references in the description of our work area, we’d like to mention that there are few academic analyses of this neighbourhood. Many of the statements come from our work for several years in the neighbourhood, the diagnosis we made to include it in the IN-HABIT research project, and the deep implication of the researchers in its reality. For this reason, we have now included two references from our work in the project.

Cruz-Piedrahita, C., Martinez-Carranza, F.-J., & Delgado-Serrano, M. M. (2024). A Multidimensional Approach to Understanding Food Deserts in Vulnerable Contexts. Sustainability, 16(3), 1136.

Delgado-Serrano, M. M., Mac Fadden, I., Martinez-Carranza, F.J., & Vancea, M (2022). Inclusive Transformation Plan of Las Palmeras. IN-HABIT Project Report 1.1. Universidad de Córdoba.

Additionally, we have strengthened the paragraph with another reference, including the sentence: “The neighbourhood faces what (Hernández Aja et al., 2014) consider cyclical vulnerability exacerbated by the absence of integrated perspectives in social policies and reliance on fragmented approaches to social resources and subsidies”. (lines 250-253)

Hernández Aja, A., Alguacil Gómez, J., & Camacho Gutiérrez, J. (2014). La vulnerabilidad urbana en España. Identificación y evolución de los barrios vulnerables. Empiria. Revista de metodología de ciencias sociales, (27), 73-94.

Reviewer 3 Report

Comments and Suggestions for Authors

Thank you for the opportunity to review this paper. I really enjoyed reading it, as I believe this focus on fusing top-down and bottom-up understandings of health and wellbeing is critical to generating sustainable urban environments. The paper included some great information and was very well written.

It would be worth including the country name as well as the city, for overseas readers to readily locate your study city.

Introduction:

Final paragraph: "The results will 125 enhance the understanding of how peripheral SMSCs work in practice. - What do you mean by 'peripheral'?

Final paragraph: "where the project will co-deploy different VIS" - What is VIS?

L:it Review: You provide good information about health and wellbeing frameworks but it would be really useful if you could incorporate detail about other studies which have attempted to do similar work (i.e. work with marginalised communities to co-create health and wellbeing indicators for urban environments) in different contexts. This information would allow your readers to appreciate how your framework might build on this existing work, and help us understand how your framework might be replicable in other contexts. 

Methods:

Study area: You mention there have been attempts at "inclusion strategies" - it would be good to know more about these past initiatives. When/How long ago were they enacted? Who created, managed and financed them? How did the locals react? etc.

How did you initially recruit your participants? e.g. via researchers' networks, public outreach campaign

You often use the present perfect to describe your methods e.g. "Three rounds of coding have 287 been done". - Does this mean the project is continuing? If so, make this clear in the Intro and Methods.

What is GDEI?

Discussion/Conclusion: It would be good to more explicitly address the issue of how your indicators might be 'measured', especially given that many of your indicators are qualitative or subjective in nature.  

Author Response

Dear Reviewer

Thanks for your kind comments and your appreciation for our work. We are very pleased by your comments

It would be worth including the country name as well as the city, for overseas readers to readily locate your study city.

Response: We have included it in the Abstract (line 15) and the Introduction (line 106).

Final paragraph: "The results will 125 enhance the understanding of how peripheral SMSCs work in practice. - What do you mean by 'peripheral'?

Response: The 4 cities of the project are located in the periphery of the European Union and are affected by the double vulnerability of not being big cities nor located in the centre of Europe. We have clarified it in lines 119-120

Final paragraph: "where the project will co-deploy different VIS" - What is VIS?

Response: We meant visionary and integrated solutions, as it is the term we use in the project (see line 118). We had already explained the meaning of VIS in line 117.

L:it Review: You provide good information about health and wellbeing frameworks but it would be really useful if you could incorporate detail about other studies which have attempted to do similar work (i.e. work with marginalised communities to co-create health and wellbeing indicators for urban environments) in different contexts. This information would allow your readers to appreciate how your framework might build on this existing work, and help us understand how your framework might be replicable in other contexts.

Response: We appreciate this comment, which has enhanced the depth of the manuscript’s literature review. To address your suggestion, we have included additional works that delve into the co-creation frameworks of indicators. Furthermore, thanks to this comment, we came across a just-published paper that aligns closely with our approach (Rohrbein et al., 2023). It employs co-creation methods to develop a system of health and wellbeing indicators at the neighbourhood level. Although their emphasis differs from our approach -they do not focus on vulnerable neighbourhoods, and they work with the health authorities, emphasising indicators of physical health-, the paper reinforces the need for co-created and context-specific indicators at the neighbourhood level. To address your comment, we have further elaborated on these ideas and introduced the following paragraph (lines 134-150): "Other researchers have previously suggested co-created frameworks and participatory approaches for assessing health and wellbeing [30,31], with a focus on physical health [32,33], or examining the use of public spaces for promoting health and wellbeing [34]. While [29] proposed the development of indicators at a neighbourhood level, their emphasis was not specifically on vulnerable neighbourhoods. Additionally, their work primarily involved health authorities and concentrated more on physical health indicators. The novelty of our approach lies in the co-creation of health and wellbeing concepts as inclusive, complex, multidimensional, and context-specific and in the integration of top-down indicators with bottom-up views in the design of health and wellbeing indicators for vulnerable contexts, using gender, diversity, equity, and inclusion perspective. This approach aims to capture the nuanced and varied experiences of individuals and communities in vulnerable settings, recognising that wellbeing encompasses more than just physical health. By incorporating diverse voices and perspectives, our framework offers a more holistic understanding of health and wellbeing in these contexts, providing valuable insights for targeted interventions and policy development."

  1. Röhrbein, H.; Hilger-Kolb, J.; Heinrich, K.; Kairies, H.; Hoffmann, K. An Iterative, Participatory Ap-proach to Developing a Neighborhood-Level Indicator System of Health and Wellbeing. Int J Environ Res Public Health 2023, 20, 1456, doi:10.3390/ijerph20021456.
  2. Slattery, P.; Saeri, A.K.; Bragge, P. Research Co-Design in Health: A Rapid Overview of Reviews. Health Res Policy Syst 2020, 18, 17, doi:10.1186/s12961-020-0528-9.
  3. Leask, C.F.; Sandlund, M.; Skelton, D.A.; Altenburg, T.M.; Cardon, G.; Chinapaw, M.J.M.; De Bourdeaudhuij, I.; Verloigne, M.; Chastin, S.F.M. Framework, Principles and Recommendations for Utilising Participatory Methodologies in the Co-Creation and Evaluation of Public Health Interventions. Res Involv Engagem 2019, 5, 2, doi:10.1186/s40900-018-0136-9.
  4. Frow, P.; McColl-Kennedy, J.R.; Payne, A. Co-Creation Practices: Their Role in Shaping a Health Care Ecosystem. Ind Mark Manag 2016, 56, 24–39, doi:10.1016/j.indmarman.2016.03.007.
  5. Bermúdez Tamayo, C.; Olry de Labry Lima, A.; García Mochón, L. Identificación de Indicadores de Buenas Prácticas En Gestión Clínica y Sanitaria. J Healthc Qual Res 2018, 33, 109–118, doi:10.1016/j.cali.2017.12.008.
  6. Villanueva, K.; Badland, H.; Hooper, P.; Koohsari, M.J.; Mavoa, S.; Davern, M.; Roberts, R.; Goldfeld, S.; Giles-Corti, B. Developing Indicators of Public Open Space to Promote Health and Wellbeing in Communities. Applied Geography 2015, 57, 112–119, doi:10.1016/j.apgeog.2014.12.003.

Methods:

Study area: You mention there have been attempts at "inclusion strategies" - it would be good to know more about these past initiatives. When/How long ago were they enacted? Who created, managed and financed them? How did the locals react? etc.

Response: Over the years, there have been many attempts in the neighbourhood, according to the participant’s comments, but we haven’t documented them because it is not easy to keep track as there are no official records. We have included some examples in lines 246-247

How did you initially recruit your participants? e.g. via researchers' networks, public outreach campaign

Response: Thanks for highlighting that these aspects were not clear enough in our manuscript. We have included the following text to address it: “A targeted engagement strategy was employed, which involved participant observation and informal discussions with residents and representatives from various public and private institutions to identify potential participants. Our initial efforts aimed to identify key formal and informal entities within Las Palmeras, such as social services, associations, NGOs, sports clubs, and religious charities. Subsequently, our focus shifted towards the most vulnerable segments of the community. To gain a comprehensive understanding, we conducted extensive research on minority and marginalised groups, including door-to-door visits to all 700 apartments in the neighbourhood, most of which attended us and helped to better understand the reality of the inhabitants”. Lines 279-287

You often use the present perfect to describe your methods e.g. "Three rounds of coding have 287 been done". - Does this mean the project is continuing? If so, make this clear in the Intro and Methods.

Response: We have changed the tense to clarify that the work has been finished

What is GDEI?

Response: GDEI stands for gender, diversity, equity and inclusion, as mentioned in line 142

Discussion/Conclusion: It would be good to more explicitly address the issue of how your indicators might be 'measured', especially given that many of your indicators are qualitative or subjective in nature. 

Response: As mentioned in lines 423-424 we are mainly measuring these indicators through surveys and questionnaires. We have already developed a baseline for it in 2022 and plan to monitor the evolution in 2025. Additionally, we have added a clarification in lines 572-574.